# Non-parallel Accent Transfer based on Fine-grained Controllable Accent Modeling

**Linqin Wang[1,2], Zhengtao Yu[1,2] *, Yuanzhang Yang[1,2],**
**Shengxiang Gao[1,2], Cunli Mao[1,2], Yuxin Huang[1,2]**
[1] Faculty of Information Engineering and Automation,
Kunming University of Science and Technology, Kunming, China
[2] Yunnan Key Laboratory of Artificial Intelligence, Kunming, China
{linqinwang7767}@163.com, {ztyu,gaoshengxiang.yn}@hotmail.com,
{maocunli,yyz845935161,huangyuxin2004}@163.com

## Abstract

Existing accent transfer works rely on parallel data or speech recognition models. This paper focuses on the practical application of accent transfer and aims to implement accent transfer using non-parallel datasets. The study has encountered the challenge of speech representation disentanglement and modeling accents. In our accent modeling transfer framework, we manage to solve these problems by two proposed methods. First, we learn the suprasegmental information associated with tone to finely model the accents in terms of tone and rhythm. Second, we propose to use mutual information learning to disentangle the accent features and control the accent of the generated speech during the inference time. Experiments show that the proposed framework attains superior performance to the baseline models in terms of accentedness and audio quality.

## 1 Introduction

The accent transfer task refers to the synthesis of speech with an accent, such as British English for North Americans. Accent pronunciation is a distinctive form of expression influenced by the native language, the speaker's social group or speaking in a particular region (Loots and Niesler, 2011). In general, people find it easier to talk with others in their own accent group. The use of speech is now widely adopted, for example in the field of chatbots and film dubbing requires research on accent transfer of speech.

At present, the accent transfer task for parallel data has achieved sound research results and increasing performance. Divided according to training data, the methods are specifically divided into: (1) **Parallel corpus** of different accents of the same speaker using source and target speech content and time alignment (Finkelstein et al., 2022; Liu et al., 2022; Hida et al., 2022; Toda et al., 2007; Oyamada et al., 2017). (2) **Non-parallel corpus** of

---

* Corresponding author

multiple speakers with multiple accents using inconsistent source and target speech content (Wang et al., 2021; Zhao et al., 2018, 2019; Kaneko and Kameoka, 2017; Kaneko et al., 2019, 2020a, 2021; Finkelstein et al., 2022) used a multi-stage trained tts model to achieve transfer of North American accents, Australian accents, and British accents, and used a CHiVE-BERT pre-training model to enhance the audio effect of accent generation. Liu et al. (2022) added an accent variance adaptor to model the rhythmicity of accent variance, and also enhanced the accent generation audio by using a consistency constraint module. The use of phonetic posteriorgrams (PPG) is an essential idea in the application of non-parallel data (Wang et al., 2021; Zhao et al., 2018, 2019). Wang et al. (2021) extracted PPG from a Chinese pre-trained speech recognition model and then used them in an end-to-end speech conversion model based on adversarial learning disentangling. This approach achieved accent transfer from Chinese Mandarin to Tianjin and obtained decent results.

However, existing accent transfer works highly rely on a large amount of labelled parallel data or advanced speech recognition models. Working with an enormous amount of labeling data is always hectic, labor-intensive, and time-consuming, which is more severe for low-resource languages. This limitation hinders the wider application of accent transfer in low-resource scenarios. Hence, it is a timely question: Is it feasible to do non-parallel accent transfer task under a unified framework? It is challenging because the speech representation containing various components, including speaker timbre, accent characteristics, and linguistic content, which are difficult to disentangle, especially for non-parallel accent transfer task.

The dataset for the task in this paper can be represented as $\{S_a(A_a), S_b(A_b)\}$, where the $a$ speaker $S_a$ can only speak the accent $A_a$, and the $b$ speaker $S_b$ can only speak the accent $A_b$. The objec-

tives of this paper are to achieve respectively: (1) $S_a(A_a) \rightarrow S_b(A_a)$, where the $A_a$ accent transfer to the $S_b$ speaker without changing the linguistic content of the speech itself. (2) $S_b(A_b) \rightarrow S_a(A_b)$, where the $A_b$ accent transfer to the $S_a$ speaker without changing the linguistic content of the speech itself. (3) $S_a(A_a) \Longleftrightarrow S_b(A_b)$, two-way speaker timbre and accent transfer between the $S_a$ speaker and the $S_b$ speaker.

Following the success of mutual information learning to disentangle speaker information in the One-shot voice conversion (VC) task (Yang et al., 2022), this paper applies the non-parallel database-based voice conversion model MaskCycleGAN-VC (Kaneko et al., 2021) to a more challenging task: voice and accent joint conversion. The source speaker's accent can be converted to the target speaker's accent without changing the linguistic content of the speech. The most challenging task is to achieve effective disentangling of accent features, linguistic features, speaker timbre features and fine-grained embodied modeling of accents in a unified model architecture, and to achieve controlled and effective speaker timbre and accent transfer in the prediction phase of the model. The accurate modeling of phonetic pronunciation tones in the task of accent transfer is crucial. The contributions of this paper are as follows.

(1) For accents being difficult to model fine-grained concretely, this paper fine-grained concretely models accents in terms of phonetic intonation, rhythmic pauses and other pronunciation features. Then, an accent feature encoder is proposed, which can effectively extract accent features in the inference stage and realise accent controllability modeling.

(2) To address the problem of difficult speaker information disentangling in non-parallel data sets with different speakers with different accents, this paper proposes mutual information learning to maximize the mutual information upper bound of accent features and speech features. It can effectively disentangle the speaker features, accent features and phonetic features of speech.

(3) Experimental results show that method converts the speech up to a MOS score of 4.12. Achieving optimal results compared to baselines on the English accent transfer task for the public VCTK dataset and on the Lao accent transfer for the self-constructed Lao dataset. It significantly improves the accentedness and audio quality.

## 2 Method

In this section, we first describe our model architecture. Then we introduce the fine-grained accent modeling and adversarial mutual information learning and show how accent transfer between speaker $S_a$ and speaker $S_b$ on non-parallel datasets with different speakers and accents is achieved.

### 2.1 Architecture of the proposed model

The generator structure of the model is shown in Figure. 1. We improve the generator part of the MaskCycleGAN-VC (Kaneko et al., 2021) for specific data and application scenarios. The generator part is composed of five parts: an accent encoder $E_{ac}$, a speaker encoder $E_s$, a speech content encoder $E_c$, a speech generator $G$, the feature disentanglements $C1$ and $C2$. The model training strategy adopts a non-parallel voice conversion approach. Given a non-parallel corpus $D(x, y)$, the training mechanism involves mapping source speech $x$ to converted speech $y$ and then back to $x'$, with the primary training objective being the minimization of the mean square error between $x$ and $x'$. The baseline models training details and our model parameters please refer to Appendix 5.

### 2.2 Encoder and accent modeling

**Encoder:** The accent encoder $E_{ac}$ takes the mel-spectrogram $S$ and normalized pitch contour $P$ of the speech as inputs. The accent encoder provides global speech accent features to control the speaker's accent. We use a vector quantized variational autoencoder (VQ-VAE) (Van Den Oord et al., 2017; Polyak et al., 2021) model to learn suprasegmental representations related to tone. The speaker encoder is the same structure in (Chen et al., 2021), using Conv1D as the main structure to extract speaker features. The speech content encoder is the downsampling module in MaskCycleGAN (Kaneko et al., 2021), the rhythm encoder $E_r$ extract speech rhythm features, we have:

$$Z_{ac} = E_{ac}(E_r(S), VQ - VAE(P)),$$
$$Z_c = E_c(S), \quad (1)$$
$$Z_t = E_s(S),$$

where $Z_{ac}$ represents the accent features, $Z_c$ represents the speech content features, $Z_t$ represents the speaker features.

**Accent modeling:** Tones alteration stands out as one of the primary distinguishing features of

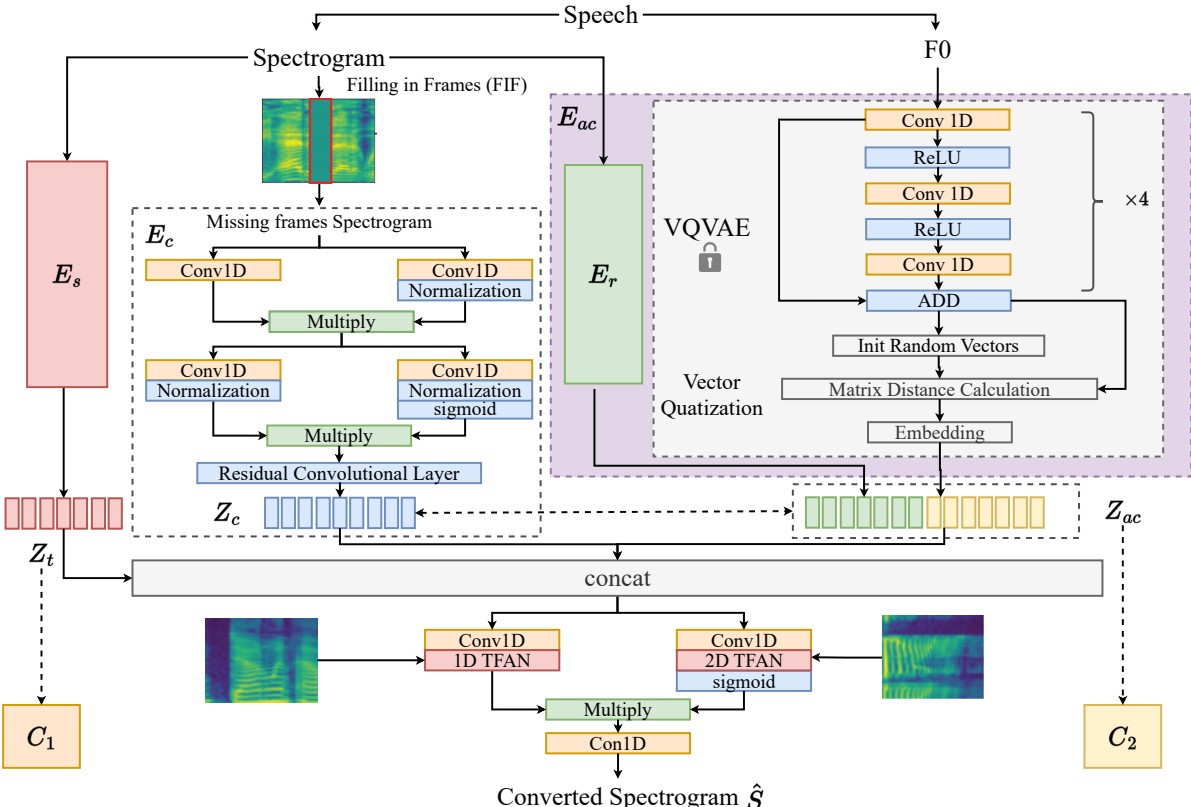

Figure 1: Framework of proposed model. The $Z_{ac}$ is the output hidden states of accent encoder $E_{ac}$, $Z_t$ is the output hidden states of speaker encoder $E_s$, $Z_c$ is the output hidden states of content encoder $E_c$, $Z_r$ is the output hidden states of rhythm encoder $E_r$. $C1$ is speaker identity classify with linear, $C2$ is accent classify with gradient reverse linear.

accents, and these different tones are reflected in the spectrum by different fundamental frequencies, so tone features can be represented by the discrete speech feature F0. To better model the tone at fine-grained degree, we adopt the VQ-VAE framework to train a pre-training model for speech tone feature F0. The yet another algorithm for pitch tracking (YAAPT) (Kasi and Zahorian, 2002) algorithm is used to extract the F0 from the input signal, $x$, generating $P = (P_1, ..., P_{T'})$, we have:

$$z_{1:L'}^F = \text{Encoder}_{\Phi_1 F}(P_{1:T'}),$$
$$e_{1:L'}^F = \text{VectorQuantization}_{\Phi_2 F}(z_{1:L'}), \quad (2)$$
$$\hat{P}_{1:L'} = \text{Decoder}_{\Phi_3}(e_{1:L'}^F),$$

Each element in $z_{1:L'}^F$ is an integer $z_s \in \{0, 1, ..., K'\}$, where $K'$ is the encoder dictionary size. The $\text{VectorQuantization}_{\Phi_2 F}$, a bottleneck with a learned codebook $C = (e_1, ..., e_{K'})$, where each item in $C$ is a 128-dimensional vector. The encoder extracts a sequence of latent vectors $\text{Encoder}_{\Phi_1 F}(P) = (h_1, ..., h_{L'})$ from the raw audio, where $h_i \in \mathbb{R}^{128}$, for all $1 \leq i \leq L'$.

Then, the bottleneck maps each latent vector to its nearest vector in the codebook $C$. The embedded latent vectors are then being fed into the decoder $\text{Decoder}_{\Phi_3}(e_{1:L'}^F) = \hat{P}$ which reconstructs the original F0 signal. Similar to (Dhariwal et al., 2020), we use Exponential Moving Average updates to learn the codebook and employ random restarts for unused embeddings, we use the indices of the mapped latent vectors to generate $Z_p$. The rhythm of the speech is extracted using the same structure as $E_r$ in SpeechSplit (Qian et al., 2020) to obtain rhythm features output $Z_r$. At last, the $Z_p \oplus Z_r$ as a representation of accent $Z_{ac}$, it is incorporated into the speech generation model. We chose $E_p$ and $E_r$ as components of the accent encoder $E_{ac}$ because pitch and rhythm represent the most significant aspects of accent variation. Additionally, recent works (Qian et al., 2020; Dhariwal et al., 2020) have demonstrated their effectiveness in extracting pitch and rhythm features. We hope that the discrete representations learned from F0 capture pitch patterns and/or other suprasegmental information. The proposed extension is straightfor-

ward, but we observe that it results in impressive improvements for accent modeling, especially F0 of reconstructed speech waveforms in Lao.

## 2.3 Generator and accent transfer

**Generator:** The speech generator $G$ is based on the architecture of MaskCycleGAN-VC (Kaneko et al., 2021) and uses the Filling in Frames (FIF) strategy during training, where a random part of the spectrogram is masked. The mel-spectrogram is downsampled and mapped from high dimension to low dimension $Z_c$. Then, the upsampled $Z_t$ (speaker features) and $Z_{ac}$ (accent features) are combined to map these features into the mel-spectrogram of the target speaker. The discriminator has the same structure as MaskCycleGAN-VC and takes as input the mel-spectrogram generated by the generator for the target speaker. we have:

$$\hat{S} = G(Z_{ac}, Z_c, Z_t), \tag{3}$$

where $\hat{S}$ represents the converted speech.

**Controllable accent transfer:** During the model inference stage, different target accents can be achieved by controlling the accent feature $Z_{ac}$.

$$\hat{S}_{S_b(A_a)} = G(Z_{ac}(A_a), Z_c(S_a), Z_t(S_b)),$$
$$\hat{S}_{S_a(A_b)} = G(Z_{ac}(A_b), Z_c(S_b), Z_t(S_a)), \tag{4}$$

where $\hat{S}_{S_b(A_a)}$ represents the speaker $S_b$ with accent $A_a$, which is not present in training data (only present the speaker $S_b$ with accent $A_b$).

## 2.4 Disentanglement and loss

**Speaker information disentanglement:** As shown in Fig. 1, we use common classifier $C1$ and adversarial speaker classifier $C2$ with gradient reverse linear (GRL) (Ganin et al., 2016) to recognize the identity of speaker, The Fig. 2 illustrates the model architecture of the two classifiers. Both classifiers, $C1$ and $C2$, utilize speaker IDs as supervisory labels during the training process. The objective of $C1$ is to accurately classify the speaker ID associated with $Z_t$ as the training progresses. On the contrary, as the training progresses, $C2$ is designed to gradually struggle in correctly classifying the speaker ID for $Z_{ac}$ and $Z_c$.

The variational contrastive log-ratio upper bound (vCLUB) (Cheng et al., 2020) is used to compute the upper bound of mutual information (MI) for irrelevant information of the speaker, decreasing

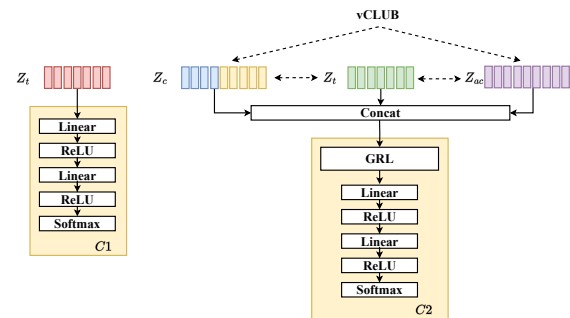

Figure 2: Framework of common classifier $C1$ and adversarial speaker classifier $C2$ .

the correlation among different speaker-irrelevant speech representations:

$$\hat{\mathcal{I}}(Z_{ac}, Z_c)_{min}$$
$$= \frac{1}{N^2} \sum_{i=1}^{N} \sum_{j=1}^{N} [\log q_\theta(Z_{ac_i}|Z_{c_i}) \tag{5}$$
$$- \log q_\theta(Z_{ac_j}|Z_{c_i})],$$

where $q_\theta(Z_{ac}|Z_c)$ is a variational distribution with parameter $\theta$ to approximate $p(Z_{ac}|Z_c)$. $\hat{\mathcal{I}}$ is the unbiased estimator for vCLUB with samples $\{Z_{ac_i}, Z_{c_i}\}$. The indexes $i$ and $j$ are the samples of $Z_{ac}$ and $Z_c$. The MI loss is:

$$\mathcal{L}_{MI} = \hat{\mathcal{I}}(Z_{ac}, Z_c). \tag{6}$$

**Loss:** The final training objective of the proposed model is to train the generator $G_{(S_a(A_a)\longleftrightarrow(S_b(A_b))}$, achieving bidirectional conversion of speech and accent between speakers $S_a$, $S_b$ and accents $A_a$, $A_b$. A full objective $\mathcal{L}_{full}$ is written as follows:

$$\mathcal{L}_{full} = \mathcal{L}_{S_a(A_a)\to S_b(A_b)}^{mask-adv}$$
$$+ \mathcal{L}_{S_b(A_b)\to S_a(A_a)}^{mask-adv}$$
$$+ \lambda_{cyc}(\mathcal{L}_{S_a(A_a)\to S_b(A_b)\to S_a(A_a)}^{cyc}$$
$$+ \mathcal{L}_{S_b(A_b)\to S_a(A_a)\to S_b(A_b)}^{cyc})$$
$$+ \lambda_{id}(\mathcal{L}_{S_a(A_a)\to S_b(A_b)}^{id} + \mathcal{L}_{S_b(A_b)\to S_a(A_a)}^{id})$$
$$+ \mathcal{L}_{S_a(A_a)\to S_b(A_b)\to S_a(A_a)}^{adv2}$$
$$+ \mathcal{L}_{S_b(A_b)\to S_a(A_a)\to S_b(A_b)}^{adv2})$$
$$+ \alpha\mathcal{L}_{com-C_1} + \beta\mathcal{L}_{adv-C_2} + \gamma\mathcal{L}_{MI}, \tag{7}$$

where $\mathcal{L}^{mask-adv}$, $\mathcal{L}^{cyc}$, $\mathcal{L}^{id}$ and $\mathcal{L}^{adv2}$ are loss function defined in MaskCycleGAN-VC (Kaneko et al., 2021). $\mathcal{L}_{com-C_1}$ and $\mathcal{L}_{adv-C_2}$ are the cross-entropy loss of the classifiers $C1$ and $C2$. The

$\mathcal{L}_{MI}$ loss corresponds to minimizing mutual information of $Z_{ac}$ and $Z_c$. $\lambda_{cyc}$, $\lambda_{id}$, $\alpha$, $\beta$ and $\gamma$ are the hyperparameters.

## 3 Experiments

### 3.1 Data

The experiments are conducted on the VCTK (Veaux et al., 2016) corpus. For English accent transfer, we conducted speaker selection involving individuals with diverse accents to validate our approach. In addition, we conducted comprehensive experiments on accent transfer in the low-resource language of Laotian. We utilize a total of 1000 Lao Vientiane accent and 1000 HuaPhan accent utterances, with 100 samples in the validation and testing set, totaling about 1 hour. All audio data used in the experiment have a sampling rate of 22.05kHz.

### 3.2 Model and training setup

In the experiments, $a$ accent $S_a(A_a)$ is used as the source speech and $b$ accent $S_b(A_b)$ is used as the target speech for training. We compare our proposed method with the current best non-parallel speech conversion models:

**CycleGAN-VC2 (Kaneko et al., 2019).** A GAN-based speech conversion model that uses mel cepstrum as input and output.

**CycleGAN-VC3 (Kaneko et al., 2020a).** This model uses mel spectrogram as input and output instead of mel cepstrum, and incorporates a time-frequency adaptive normalization (TFAN) module on the basis of CycleGAN-VC2.

**MaskCycleGAN-VC (Kaneko et al., 2021).** This model adds a mask mechanism on the basis of CycleGAN-VC2.

**SRD (Yang et al., 2022).** A method that disentangles speaker information based on mutual information learning.

In the experimental application, the feature of the speech is an 80-dimensional Mel-spectrogram, and the tone feature is represented F0. The F0 is extracted from the raw audio using a window size of $20ms$ and a $5ms$ hop. The VQ-VAE quantization described at Sec. 2.2, is applied using an F0 codebook of $K0 = 20$ tokens and an encoder that downsamples the signal by $\times 16$. The configurations of MaskCycleGAN-VC followed its original paper (Kaneko et al., 2021). We chose the model checkpoint with the lowest loss on the

Table 1: Evaluation results of different models. Average MCD, RMSE, SSIM, MOS with $95\%$ confidence between the converted speech and the ground truth reference. "*" denotes the proposed model. "w/o" is short for "without" in ablation study. The "$A_{p243}$", "$A_{p329}$", "$A_{p248}$" and "$A_{p244}$" represent the London accent, American accent, Indian accent and Manchester accent of English, respectively. $\Delta$: the difference value between our model and the best baseline model, $\uparrow$: improved performance compared to the best baseline model, **Bold**: the best performance under each category, underline: the second best performance, "–": results are not available.

| Accent Transfer | Methods | MCD | SSIM | RMSE | MOS |
|---|---|---|---|---|---|
| $S_{p243}(A_{p243})$ $\Longleftrightarrow$ $S_{p329}(A_{p329})$ | CycleGAN-VC2 | 8.16 | 0.59 | 41.38 | 2.61±0.08 |
| | CycleGAN-VC3 | 6.12 | 0.84 | 29.54 | 4.01±0.13 |
| | MaskCycleGAN-VC | 6.82 | 0.79 | 31.21 | 3.89±0.16 |
| | SRD | 6.94 | 0.83 | 30.05 | 3.83±0.11 |
| | Our Model* | **5.61** | **0.86** | **29.40** | **4.08±0.17** |
| | $\Delta$ | ↑0.51 | ↑0.02 | ↑0.14 | ↑0.07 |
| $S_{p243}(A_{p243})$ $\rightarrow$ $S_{p329}(A_{p243})$ | CycleGAN-VC2 | - | - | - | - |
| | CycleGAN-VC3 | - | - | - | - |
| | MaskCycleGAN-VC | - | - | - | - |
| | SRD | 6.79 | 0.78 | 32.56 | 3.98±0.12 |
| | Our Model* | **5.59** | **0.83** | **28.15** | **4.05±0.18** |
| | $\Delta$ | ↑1.20 | ↑0.05 | ↑4.41 | ↑0.07 |
| $S_{p329}(A_{p329})$ $\rightarrow$ $S_{p243}(A_{p329})$ | CycleGAN-VC2 | - | - | - | - |
| | CycleGAN-VC3 | - | - | - | - |
| | MaskCycleGAN-VC | - | - | - | - |
| | SRD | 6.56 | **0.79** | 35.64 | 3.85±0.12 |
| | Our Model* | **5.97** | 0.76 | **32.02** | **3.92±0.13** |
| | $\Delta$ | ↑0.59 | ↓0.03 | ↑3.62 | ↑0.07 |
| $S_{p243}(A_{p243})$ $\rightarrow$ $S_{p329}(A_{p243})$ | w/o $Z_r$* | 6.12 | 0.69 | 33.12 | 3.81±0.13 |
| | w/o $Z_{p(VQ-VAE)}$* | 7.52 | 0.52 | 36.52 | 3.23±0.17 |
| | w/o $Z_{ac}$* | 8.96 | 0.37 | 53.52 | 2.05±0.16 |
| $S_{p248}(A_{p248})$ $\Longleftrightarrow$ $S_{p244}(A_{p244})$ | CycleGAN-VC2 | 7.57 | 0.58 | 38.14 | 3.08±0.19 |
| | CycleGAN-VC3 | 5.65 | **0.85** | 33.98 | 4.01±0.18 |
| | MaskCycleGAN-VC | 5.83 | 0.69 | 35.72 | 3.86±0.13 |
| | SRD | 6.12 | 0.74 | **32.18** | 3.92±0.11 |
| | Our Model* | **5.53** | 0.83 | 34.80 | **4.02±0.15** |
| | $\Delta$ | ↑0.12 | ↓0.02 | ↓2.62 | ↑0.01 |
| $S_{p248}(A_{p248})$ $\rightarrow$ $S_{p244}(A_{p248})$ | CycleGAN-VC2 | - | - | - | - |
| | CycleGAN-VC3 | - | - | - | - |
| | MaskCycleGAN-VC | - | - | - | - |
| | SRD | 7.12 | **0.83** | **33.87** | 3.69±0.11 |
| | Our Model* | **5.69** | 0.78 | 34.49 | **3.95±0.13** |
| | $\Delta$ | ↑1.43 | ↓0.05 | ↓0.62 | ↑0.26 |
| $S_{p243}(A_{p243})$ $\Longleftrightarrow$ $S_{p248}(A_{p248})$ | CycleGAN-VC2 | 6.45 | 0.80 | 57.38 | 3.01±0.15 |
| | CycleGAN-VC3 | 5.22 | **0.87** | 42.69 | 3.82±0.11 |
| | MaskCycleGAN-VC | 5.59 | 0.76 | 49.41 | 3.98±0.11 |
| | SRD | 6.92 | 0.71 | 35.69 | 3.92±0.13 |
| | Our Model* | 6.08 | 0.79 | **30.86** | **4.08±0.12** |
| | $\Delta$ | ↓0.86 | ↓0.08 | ↑4.83 | ↑0.10 |
| $S_{p243}(A_{p243})$ $\rightarrow$ $S_{p248}(A_{p243})$ | CycleGAN-VC2 | - | - | - | - |
| | CycleGAN-VC3 | - | - | - | - |
| | CycleGAN-VC3 | - | - | - | - |
| | SRD | 7.58 | 0.60 | 36.42 | 3.85±0.13 |
| | Our Model* | **6.44** | **0.79** | **31.01** | **4.11±0.12** |
| | $\Delta$ | ↑1.44 | ↑0.19 | ↑5.41 | ↑0.26 |
| $S_{p248}(A_{p248})$ $\rightarrow$ $S_{p243}(A_{p248})$ | CycleGAN-VC2 | - | - | - | - |
| | CycleGAN-VC3 | - | - | - | - |
| | MaskCycleGAN-VC | - | - | - | - |
| | SRD | 7.10 | 0.59 | 35.50 | 3.91±0.13 |
| | Our Model* | **6.17** | **0.68** | **32.10** | **4.12±0.18** |
| | $\Delta$ | ↑0.93 | ↑0.09 | ↑3.40 | ↑0.21 |
| $S_{p243}(A_{p243})$ $\rightarrow$ $S_{p248}(A_{243})$ | w/o $Z_r$* | 7.69 | 0.43 | 29.36 | 3.90±0.13 |
| | w/o $Z_{p(VQ-VAE)}$* | 9.12 | 0.45 | 37.52 | 3.33±0.14 |
| | w/o $Z_{ac}$* | 9.79 | 0.28 | 59.52 | 2.05±0.17 |
| $S_{p248}(A_{p248})$ $\Longleftrightarrow$ $S_{p329}(A_{p329})$ | CycleGAN-VC2 | 6.76 | 0.76 | 54.39 | 3.34±0.14 |
| | CycleGAN-VC3 | **5.41** | **0.83** | **32.17** | 3.67±0.15 |
| | MaskCycleGAN-VC | 5.51 | 0.79 | 42.15 | 3.94±0.08 |
| | SRD | 5.98 | 0.73 | 35.23 | 3.82±0.15 |
| | Our Model* | 5.48 | 0.81 | 33.01 | **3.97±0.11** |
| | $\Delta$ | ↓0.07 | ↓0.02 | ↓0.84 | ↑0.03 |
| $S_{p248}(A_{p248})$ $\rightarrow$ $S_{p329}(A_{p248})$ | CycleGAN-VC2 | - | - | - | - |
| | CycleGAN-VC3 | - | - | - | - |
| | CycleGAN-VC3 | - | - | - | - |
| | SRD | 6.58 | **0.82** | 35.32 | 3.99±0.12 |
| | Our Model* | **5.68** | 0.81 | **33.58** | **4.04±0.11** |
| | $\Delta$ | ↑0.42 | ↓0.01 | ↑1.74 | ↑0.05 |

validation set. HiFiGAN vocoder ([Kong et al., 2020](#)) is employed to generate speech waveforms from mel-spectrograms. In the conversion model training stage, conversion model is trained for 100 epochs using batch size of 1. We use Adam optimizer ([Kingma and Ba, 2014](#)) with learning rate is 0.0002, $\beta_1 = 0.5$, $\beta_2 = 0.999$ for speech generator. With learning rate is 0.0001, $\beta_1 = 0.5$, $\beta_2 = 0.999$ for speech discriminator optimizer. All experiments are conducted on a single NVIDIA 3090 for training.

## 3.3 Experimental results and analysis

### 3.3.1 Accent Similarity and Speaker Similarity

In order to verify the accent similarity and speaker similarity between the converted speech and the original target speech, we perform a accent and speaker visualization using t-SNE method ([Van der Maaten and Hinton, 2008](#)) based on the accent representation $Z_{ac}$ and speaker representation $Z_t$ of different speakers utterances. For accent similarity, there are 200 utterances sampled for two speaker $(S_{p243}, S_{p248})$ to calculate the accent representation. Meanwhile, we concatenate $Z_p$ and $Z_r$ in SRD and compare it with our proposed method. As can be seen in Figure. 3(b) and Figure. 3(c), While SRD partially represents different speaker accents in terms of pitch and rhythm, most accent hiiden states are still mixed together. In contrast, the accent embeddings in our approach are separable for different speakers, with only a few representations being slightly mixed. Our analysis of the Vientiane and Huaphan accents in Lao shows some similarity in certain sentences. These results indicates that our fine-grained Lao accent modeling encoder $E_{ac}$ is capable of extracting the accent $Z_{ac}$ as speaker accent information.

For speaker similarity, We randomly selecte 200 utterances and converted them to 2 target speakers $(S_w, S_h)$ each with 2 accents $(A_w, A_h)$ in Lao. The speacker representation is calculated using the speaker encoder. Subsequently, we visualize the speaker representation $Z_t$ by t-SNE ([Van der Maaten and Hinton, 2008](#)) in Figure. 3(a). The results demonstrate that all samples were grouped into two clusters representing the two target speakers. This unveils that the output speech samples from our model, including those converted samples with non-native new accent, have successfully preserved the speaker similarity of the target speakers.

Table 2: Evaluation results of different models. Average MCD, RMSE, SSIM, MOS with $95\%$ confidence between the converted speech and the ground truth reference. "*" denotes the proposed model. "w/o" is short for "without" in ablation study. The "$A_w$" is Vientiane accent of Lao, "$A_h$" is Huaphan accent of Lao. $\Delta$: the difference value between our model and the best baseline model, $\uparrow$: improved performance compared to the best baseline model, **Bold**: the best performance under each category, underline: the second best performance, "–": results are not available.

| Accent Transfer | Methods | MCD | SSIM | RMSE | MOS |
|---|---|---|---|---|---|
| $S_w(A_w)$ $\iff$ $S_h(A_h)$ | CycleGAN-VC2 | 8.49 | 0.68 | 40.70 | 2.82±0.17 |
| | CycleGAN-VC3 | 7.17 | 0.81 | **29.44** | 3.95±0.15 |
| | MaskCycleGAN-VC | 7.65 | 0.67 | 35.61 | 3.79±0.15 |
| | SRD | 7.70 | 0.83 | 33.80 | 3.85±0.12 |
| | Our Model* | **7.09** | **0.85** | 29.87 | **4.01±0.16** |
| | $\Delta$ | ↑0.08 | ↑0.02 | ↓0.43 | ↑0.06 |
| $S_w(A_w)$ $\rightarrow$ $S_h(A_w)$ | CycleGAN-VC2 | - | - | - | - |
| | CycleGAN-VC3 | - | - | - | - |
| | MaskCycleGAN-VC | - | - | - | - |
| | SRD | 7.56 | 0.81 | 34.73 | 3.73±0.12 |
| | **Our Model*** | **7.32** | **0.85** | **31.01** | **3.98±0.14** |
| | $\Delta$ | ↑0.24 | ↑0.04 | ↑3.72 | ↑0.25 |
| $S_h(A_h)$ $\rightarrow$ $S_w(A_h)$ | CycleGAN-VC2 | - | - | - | - |
| | CycleGAN-VC3 | - | - | - | - |
| | MaskCycleGAN-VC | - | - | - | - |
| | SRD | 7.58 | 0.40 | 35.64 | 3.85±0.12 |
| | **Our Model*** | **7.16** | **0.57** | **30.02** | **4.02±0.16** |
| | $\Delta$ | ↑0.42 | ↑0.17 | ↑5.62 | ↑0.17 |
| $S_w(A_w)$ $\rightarrow$ $S_h(A_w)$ | w/o $Z_r$* | 7.92 | 0.49 | 30.36 | 3.79±0.12 |
| | w/o $Z_{p(VQ-VAE)}$* | 8.62 | 0.36 | 35.52 | 3.53±0.16 |
| | w/o $Z_{ac}$* | 9.88 | 0.24 | 65.52 | 1.95±0.17 |

## 3.3.2 Objective Evaluation

For objective evaluation, we use mel-cepstrum distortion (MCD) ([Toda et al., 2007](#)), root mean square errors (RMSE) ([Luo et al., 2017](#)) between synthesised and reference speech utterances. The lower the MCD is, the smaller the distortion, meaning that the two audio segments are more similar to each other. To evaluate intonation variations of the converted voice, RMSE of source and converted voice is calculated. To account for the temporal difference, dynamic time warping is performed between the converted utterance and the target reference to compute MCD and RMSE, where the RMSE of F0 is calculated only on the voiced frames in the reference utterances. To evaluate the proposed method objectively, 50 conversion utterances pair are randomly selected. Table 1 summarizes the MCD, SSIM ([Wang et al., 2004](#)) and RMSE evaluation results on VCTK datasets. It is worth noting that in the accent transfer tasks $(S_{p243}(A_{p243}) \rightarrow S_{p248}(A_{p243})$, $S_{p248}(A_{p248}) \rightarrow S_{p243}(A_{p248}))$, we do not have data on real labels, so we did a cycle convert, e.g., $S_{p243}(A_{p243}) \rightarrow S_{p248}(A_{p243}) \rightarrow \tilde{S}_{p243}(A_{p243})$. It is observed that: our model outperforms all baseline models consistently for MCD and achieves the best

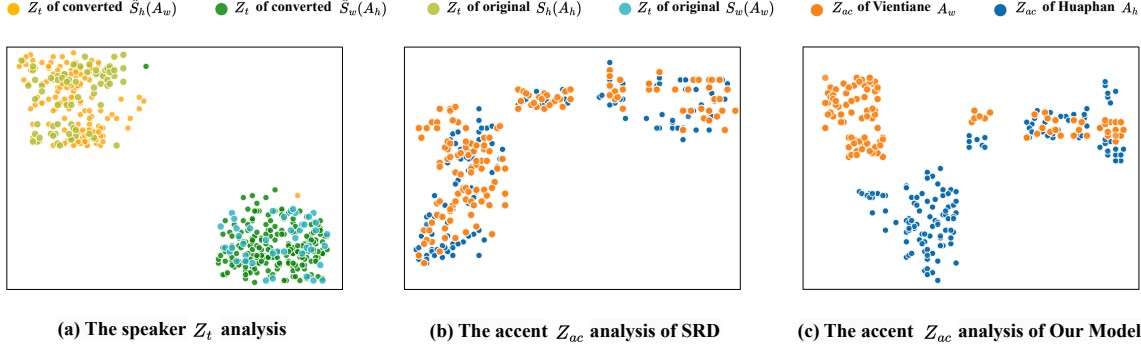

Figure 3: The different encoder output visualization using t-SNE. (a) $E_s$ encoder output $Z_t$ visualization of our model, (b) $E_{ac}$ encoder output $Z_{ac}$ visualization of SRD baseline, (c) $E_{ac}$ encoder output $Z_{ac}$ visualization of our model.

or second best results of SSIM, which shows the proposed method has better intelligibility while preserving the source linguistic content. In addition, in Table 1, we note that our model achieves the lowest RMSE in the task of English accent transfer involving four distinct accents, which shows the ability of our model in transforming and preserving the detailed intonation variations from source speech to the converted one. This indicates the effectiveness of the proposed fine-grained lao accent modeling encoder. Similar conclusions are obtained in the transfer of the low-resource Lao between Vientiane accents ($A_w$) and Huaphan accents ($A_h$) as shown in Table 2.

### 3.3.3 Subjective Evaluation

Subjective evaluations are conducted using listening tests with human subjects. AB preference test is performed to evaluate speech quality and speaker similarity, respectively. Additionally, mean opinion score (MOS) tests are conducted to determine listeners' preferences across all experimental methods. For each test, 20 samples were randomly selected from the converted samples of each experimental system and provided to 15 participants.

**Audio Quality:** In the MOS test, listeners are asked to rate the quality of the converted speech on a 5-point scale. Audios converted from the three systems are randomly shuffled before presenting to listeners. Each group of audio corresponds to the same text content. The MOS results in Table 1 show that our model achieves the best quality MOS as compared with other methods on the VCTK dataset. Simultaneously, better performance has been achieved in the task of accent transfer involving English British (p243) and American (p329)

accents, as well as Indian (p248) and British (p243) accents, and Indian (p248) and American (p329) accents. In addition, the MOS results in Table 1 show that our model also performs well in the low-resource language Lao of both $S_w(A_w) \rightarrow S_h(A_w)$, $S_h(A_h) \rightarrow S_w(A_h)$, $S_w(A_w) \Longleftrightarrow S_h(A_h)$ accent transfer tasks. Note that the baseline methods (MaskCycleGAN, CycleGAN-VC3, CycleGAN-VC2) cannot generate samples properly with $a$ accent ($A_a$) and $b$ accent ($A_b$). Because these non-parallel speech conversion models do not have the ability to disentangle speaker representations and accent representations. Hence these mothods do not have samples for MOS test. SRD has the ability to decouple speech features such as rhythm, pitch, and content, which enables it to achieve a certain degree of accent conversion. However, due to the lack of modeling of the tone by pre-trained VQ-VAE models, its performance in accent conversion tasks is not as good as our proposed method.

**Accentedness:** In the AB test on accentedness, paired speech samples with the same textual content are presented and the listeners are asked to choose samples that are more similar to the target accent. The results are shown in Figure. 4. Irrespective of the language, the proposed model is more effective with much more preference.

### 3.3.4 Conversion Visualization

Lao language is a tonal language where the variations in tones become more pronounced with changes in intonation (Erickson, 2001). Figure. 5 shows the spectrogram and F0 of the source $S_w(A_w)$, target $S_h(A_h)$ and converted speeches $\hat{S}_h(A_w)$ with the same Lao content. Please note we use parallel speech data to visualize the re-

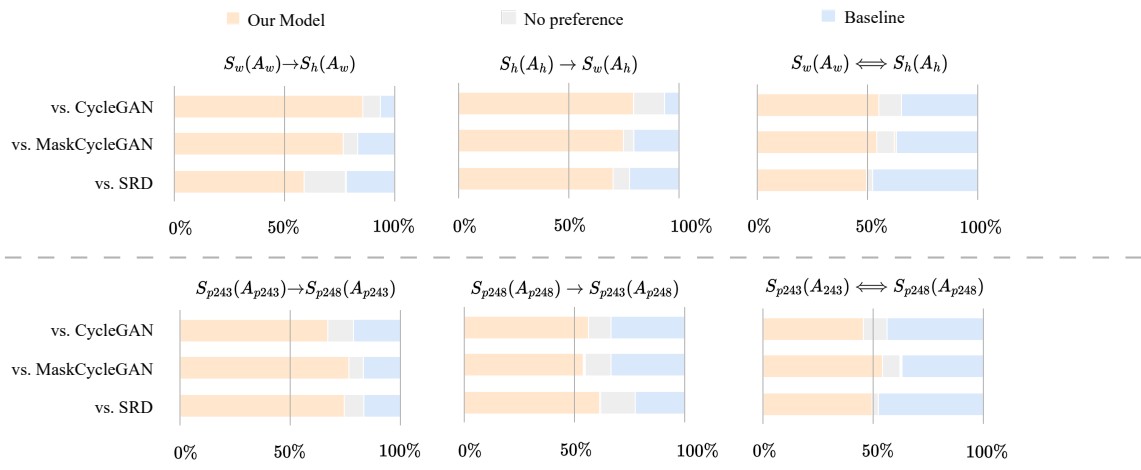

Figure 4: Accentedness preference test results



(a) Spectrogram and F0 of source $S_w(A_w)$  (b) Spectrogram and F0 of converted $\hat{S}_h(A_w)$  (c) Spectrogram and F0 of target $S_h(A_h)$

Figure 5: The comparison of spectrogram and F0 for source $S_w(A_w)$, target $S_h(A_h)$ and converted speeches $\hat{S}_h(A_w)$. Horizontal axis (x-axis) displays time in second, and vertical axis (y-axis) represents spectral frequency and F0 frequency respectively

sults. For accent conversion $S_w(A_w){\rightarrow}S_h(A_w)$, the F0 contour of the converted speech matches average pitch of the source speech and retains detailed characteristics of the source pitch contour. The spectrogram details of the converted speech match of the target speech, but there are significant differences in the contour of the F0 pitch. The results demonstrated that our model has successfully achieved accent conversion on non-parallel training data $\{S_w(A_w), S_h(A_h)\}$.

### 3.3.5  Ablation study

Moreover, we conduct ablation study that addresses performance effects from different methods for lao accent modeling with results shown in the last three rows of Table 1. From the results, when the lao accent modeling $Z_{ac}$ without the $Z_r$ of speech rhythm, the model is still able to perform accent transfer and outperforms most of the baseline models, but the speech naturalness decrease. When the VQ-VAE framework of F0 modeling is removed, the audio quality and accentedness significant decrease. When the lao accent modeling $Z_{ac}$ is removed, the results are poor and no longer perform

the accent transfer task well.

## 4  Conclusions

Based on the application scenario of accent transfer in non-parallel data sets, this paper proposes a non-parallel accent transfer method based on fine-grained controllable accent modeling. It applies a VQ-VAE network for fine-grained modeling of voice intonation and rhythmic pauses, and then delivers the obtained accent features and speech features to a mutual information-based learning feature disentangler. The features extracted by the trained accent encoder can guide the pitch and rhythm variation of the generated speech in the prediction stage of the converted model, achieving controllable modeling of the various accents. The proposed method generates speech with greatly improved fluency and naturalness, and achieves accent transfer in non-parallel dataset through the application of a unified speech conversion framework.

## Acknowledgements

This work was supported in part by the National Natural Science Foundation of China (Nos. 62376111, U21B2027 and 61972186), Yunnan provincial major science and technology special plan projects (Nos. 202103AA080015 and 202302AD080003), Yunnan Provincial Key Research and Development Plan (Nos. 202303AP140008). The authors would like to thank anonymous reviewers for their comments.

## Limitations

In this paper, we have focused on modeling accents by examining pitch and rhythmic changes in speech. However, in our future work, we plan to analyze accents by incorporating additional features of speech. By doing so, we aim to enhance the authenticity of accent performance and improve our understanding of accent variations.

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

# 5  Appendix: CycleGANs

CycleGANVC/VC2/VC3/Mask (Kaneko et al., 2019, 2020a,b, 2021) (CycleGANS) is a voice conversion model consisting of two generators, $G$, and two discriminators, $D$. CycleGANs has emerged as a novel approach in the domain of voice conversion, demonstrating its effectiveness in learning the transformation between different acoustic feature sequences without the need for parallel data. These advancements contribute to the ongoing progress in voice conversion research and its applications in various fields. The primary objective of CycleGANs, as discussed in the research papers by Kaneko and Kameoka (Kaneko et al., 2020a,b, 2021), is to acquire the ability to transform acoustic feature sequences belonging to the source domain $X$ into those of the target domain $Y$ without relying on parallel data. The acoustic feature sequences are represented by $x \in R^{Q \times T}$ and $y \in R^{Q \times T}$, where $Q$ and $T$ represents the feature dimension and the sequence length respectively.

The foundation of CycleGANs is rooted in the inspiration drawn from CycleGAN, originally proposed for image-to-image style transfer in computer vision. By applying the principles of CycleGAN, CycleGANs aims to learn the mapping function $G(x) \to Y$, enabling the conversion of input $x \in X$ to output $y \in Y$.

In pursuit of this goal, CycleGAN semploys several loss functions during the learning process. These include adversarial loss, cyclic consistency loss, and identity mapping loss, collectively contributing to the enhancement of the quality and fidelity of the generated outputs. CycleGAN-VC2 (Kaneko et al., 2020a) introduces an additional adversarial loss to further refine and improve the fine-grained details of the reconstructed features. CycleGAN-VC3 (Kaneko et al., 2020b) incorporats an additional module called time-frequency adaptive normalization (TFAN). Although the performance is superior, an increase in the number of converter parameters is necessary (from 16M to 27M). MASKCycleGAN (Kaneko et al., 2021) use a novel auxiliary task called filling in frames (FIF), which apply a temporal mask to

Table 3: Forms and interpretations of notations.

| Symbol | Definition |
|---|---|
| $x$ | Original data of speech $a$ |
| $y$ | Original data of speech $b$ |
| $x'$ | Generate new sample of speech $a$ |
| $y'$ | Generate new sample of speech $b$ |
| $G_{\theta_1}^{X \to Y}$ | Forward conversion from speech $a$ to speech $b$ with parameters $\theta_1$ |
| $G_{\theta_2}^{Y \to X}$ | Inverse conversion from speech $b$ to speech $a$ with parameters $\theta_2$ |
| $S$ | The mel-spectrogram |
| $P$ | The normalized pitch contour |
| $S_a$ | The speaker identity of speech $a$ |
| $S_b$ | The speaker identity of speech $b$ |
| $A_a$ | The accent identity of speech $a$ |
| $A_b$ | The accent identity of speech $b$ |
| $E_{ac}$ | An accent encoder |
| $E_s$ | A speaker encoder |
| $E_c$ | A speech content encoder |
| $C_1$ | A speaker identity classifier with linear |
| $C_2$ | A speaker identity classifier with gradient reverse linear |
| $Z_{ac}$ | The accent feature from $E_{ac}$ |
| $Z_t$ | The speaker feature from $E_s$ |
| $Z_c$ | The speaker feature from $E_c$ |
| $Z_{ac}$ | The accent feature from $E_{ac}$ |
| $\hat{\mathcal{I}}$ | The unbiased estimator for vCLUB (Cheng et al., 2020) |

the input mel-spectrogram and encourage the converter to fill in missing frames based on surrounding frames. These adjustments add some structure to the text and make it even more reader-friendly.

This paper applies the non-parallel data-based voice conversion model MaskCycleGAN-VC (Kaneko et al., 2021) to a more challenging task: voice and accent joint conversion. The source speaker's accent can be converted to the target speaker's accent without changing the linguistic content of the speech. We improve the generator part of the MaskCycleGAN-VC (Kaneko et al., 2021) for specific data and application scenarios. The comprehensive list of the primary symbols used throughout this paper is presented in Table 3.