# OpenReview forum: "Non-parallel Accent Transfer based on Fine-grained Controllable Accent Modelling"
_EMNLP/2023/Conference — EMNLP 2023 Findings_

### Official Review · Reviewer_sM6F · 2023-07-27

**Soundness:** 3

**Excitement:**

3: Ambivalent: It has merits (e.g., it reports state-of-the-art results, the idea is nice), but there are key weaknesses (e.g., it describes incremental work), and it can significantly benefit from another round of revision. However, I won't object to accepting it if my co-reviewers champion it.

**Paper Topic And Main Contributions:**

This paper proposes a framework that can do both accent and voice conversion without parallel data (I assume that means paired data with the same speech content but different accent etc.). They achieved this by disentangling the latent representations.

**Questions For The Authors:**

In addition to RR:
1. What is MI? This is not defined at all.
2. Loads of terms are not defined in Eqn. 5.

**Reasons To Accept:**

1. The proposed method shows good performance with adequate analysis

**Reasons To Reject:**

1. There are some technical flaws in this paper: The definition of C1 is speaker classification, and it takes Z_ac as input which is meant to capture accent information (which should be speaker independent?). Also C2 takes the concatenated Z as input but was trained in an adversarial way - isn't it going to confuse Z_ac what it should capture exactly?
2. The writing of this paper is poor: It makes grammatically mistakes from the very first sentence (works-> work). The nomenclature is weird: e.g. it uses C as the codebook and then C1 and C2 as two tasks.


**Reproducibility:**

4: Could mostly reproduce the results, but there may be some variation because of sample variance or minor variations in their interpretation of the protocol or method.

**Reviewer Confidence:**

5: Positive that my evaluation is correct. I read the paper very carefully and I am very familiar with related work.

---

> ### Author Rebuttal · Authors · 2023-08-28
>
> (1)Both classifiers, C1 and C2, utilize speaker IDs as supervisory labels during the training process. The objective of C1 is to accurately classify the speaker ID associated with Z_t as the training progresses. On the contrary, as the training progresses, C2 is designed to gradually struggle in correctly classifying the speaker ID for Z_{ac} and Z_c. The input of C1 is Z_t, which is trained to proficiently differentiate between various speaker intonations. The input of C2 is Z_ac, which is trained in an adversarial way to decouple Z_ac and Z_t, transforming accent features into speaker-independent vectors.
> (2)MI is mutual information, .Here is a detailed explanation of each L loss function of Eqn. 5
> $\mathcal{L}_{cyc}^{X\longrightarrow Y\longrightarrow X} =\mathbb{E}_x~P_X[||G_{Y\to X}(G_{X\to Y}(x))-x||_1]$,
> $\mathcal{L}_{id}^{X\longrightarrow Y} =\mathbb{E}_{y\sim P_Y}[||G_{X\to Y}(y)-y||_1]$,
>  \mathcal{L}_{adv2}^{X\longrightarrow Y\longrightarrow X} =\mathbb{E}_{x\sim P_X}[\log_{}{D'_X}(x)]+\mathbb{E}_{x\sim P_X}[\log_{}{(1-D'_X}(G_{Y\to X}(G_{X\to Y}(x))))], \mathcal{L}_{com-C_1}}  and {\mathcal{L}_{adv-C_2} are the cross-entropy loss of the classifiers C1 and C2. The \mathcal{L}_{MI} loss corresponds to minimizing mutual information of Z_ac and Z_c.

---

### Official Review · Reviewer_4Wqo · 2023-07-30

**Soundness:** 3

**Excitement:**

3: Ambivalent: It has merits (e.g., it reports state-of-the-art results, the idea is nice), but there are key weaknesses (e.g., it describes incremental work), and it can significantly benefit from another round of revision. However, I won't object to accepting it if my co-reviewers champion it.

**Paper Topic And Main Contributions:**

The paper proposes a system to model the task of non-parallel accent transfer and achieves promising results. Concretely, the system does two things:
1. Propose an architecture to create accent embeddings given speech (suprasegmental features) as an input (Z_ac, in the paper)
2. Disentangle speaker information and accent information, thereby allowing for a non-parallel accent transfer of a particular speaker. (Z_t, in the paper)

Use above two, along with existing method to capture linguistic content (Z_c, in the paper), to generate accented speech using MaskCycleGaN.

The results are shown in two languages, i.e., English and Laos. The proposed model achieves the best results in terms of Mean Opinion Score (MOS), MCD, SSIM, RMSE compared to other systems that model the same task.

**Questions For The Authors:**

- [A] How do these models compare against parallel accent transfer techniques? While I understand that they would do much better, it would be nice to see the comparisons (how close we are to these techniques).
- [B] How many iterations of experiments have you run? Would like to understand how stable the proposed model is.
- [C] In Figure 1, what does the arrow joining Z_c and Z_ac indicate?

**Reasons To Accept:**

- Applications to the research work done (non-parallel accent transfer) are numerous and the topic in general is exciting.
- The results seem promising, achieving SOTA results in the non-parallel accent transfer domain.
- It seems to apply for different kinds of languages — English (high-resource, non-tonal) and Laos (low-resource, tonal).
- Excellent ablation studies to show if all the features used for modelling are truly needed for performance improvement (and also how much each of the features contributes to the performance).
- Great visualisations to show how well the accent transfers (t-SNE plots) if the proposed model is applied.

**Reasons To Reject:**

- I found the paper difficult to follow — possibly because of the way it’s presented.  E_r and E_s seem to be integral part of the network proposed. However, the motivation to choose those specific architectures for E_s and E_r are unclear (at least directly from the paper). Additionally, one has to refer to the papers to understand them (keeping in mind that this is for a wider NLP audience, who are not necessarily up to date with VC task). Instead it would be nice to briefly explain them in the paper (or in the appendix) so that its easy to refer.
- Again keeping in mind that this is a venue for wider NLP audience, it would be nice to have a section in the appendix explaining what MaskCycleGAN-VC is.
- The parameters do seem underspecified to me. Would be nice to have them all in the appendix for easy reproducibility.

**Reproducibility:**

3: Could reproduce the results with some difficulty. The settings of parameters are underspecified or subjectively determined; the training/evaluation data are not widely available.

**Reviewer Confidence:**

2: Willing to defend my evaluation, but it is fairly likely that I missed some details, didn't understand some central points, or can't be sure about the novelty of the work.

**Typos Grammar Style And Presentation Improvements:**

The paper would, in general, benefit from another round of proof-reading. Here is a non-exhaustive list of typos and possible fixes (this did not affect the ratings)
- Zr in line 203 not defined? Er could be defined better.
- Equation (3), is Zs supposed to be Zt? Zs and Zt are used interchangeably following this equation (such as Equation (4) as well. Would be nice if the authors fix all such occurrences so that its easier for readers to follow the work.
- Defining abbreviations — GRL, MI, vCLUB, TFAN, SRD, YAAPT etc.
- Line 030, “people find it easier”
- Line 074, “timing question” -> timely question? I am unsure if thats the authors meant.
- Line 079-081, “which are difficult .. transfer task” doesn’t read well to me. Instead, “. These fine-grained features are difficult to disentangle, especially for non-parallel accent transfer task”
- Line 116-117, “this paper fine-grained concretely models..” -> “this paper fine-grained concretely models..”
- Line 119, “a accent” -> “an accent”
- Line 148, “we” -> “We”
- Line 160-163, “which using .. related to tone” -> “. We use a VQ-VAE (Van Den et. al.) model to learn suprasegmental representations related to tone”
- Line 191, “row” -> “raw”
- Line 332, “most accent hiidden” -> “most accent hidden”
- Line 358, “root mean square of RMSE”, should be “root mean square of ??”
- Line 363, “F0RMSE” -> RMSE of F0?
- Line 382-383, “with preserve the source linguistic content” -> while preserving the source linguistic content
- Line 383, “We” -> “we”
- Line 383, “our model achieve” -> “our model achieves”
- Line 384, “In” -> “in”
- Line 383-390, “In addition..modelling encoder” is difficult read. Better to paraphrase/split them into multiple sentences
- Line 426, “distangle” -> “disentangle”
- Line 440, “mater” -> “matter”
- I see modelling (Line 432) and modeling (Title, Line 172), would be nice to follow one throughout the paper.
- Line 440-441, “No mater the language is English or Lao” -> “Irrespective of the language”
- Line 442, “are” -> “is”

---

> ### Author Rebuttal · Authors · 2023-08-28
>
> (1). The E_r is used to extracte the rhythm of the speech, which has the same structure in SpeechSplit\cite{qian2020unsupervised}. The E_t is the speaker encoder to extract speaker features, which is the same structure in \cite{chen2021again}. Our selection of these architectures was influenced by their demonstrated capability to effectively extract both speaker timbre features and speech rhythm features in prior research.
> (2). We will provide more comprehensive parameter descriptions pertaining to this paper, along with an elaboration on the workings of MaskcycleGan, within the appendix section.
> (3).
> QUESTION A: How do these models compare against parallel accent transfer techniques? While I understand that they would do much better, it would be nice to see the comparisons (how close we are to these techniques).
> ANSWER A: The MOS score of parallel accent transfer techniques is 4.57, closely aligned with the reference waveform score of 4.59\citet{liu2022controllable},  indicating a remarkably high quality of generated speech that closely approximates actual labeled data. Our model achieves a MOS score of 4.02, implying a slight disparity compared to the reference label.
> QUESTION B: How many iterations of experiments have you run? Would like to understand how stable the proposed model is.
> ANSWER B: We conduct each experiment 5 times and report the mean score.
> QUESTION C: In Figure 1, what does the arrow joining Z_c and Z_ac indicate?
> ANSWER C: It indicates the Mutual information disentanglement learning between Z_c and Z_ac.

---

### Official Review · Reviewer_hXtk · 2023-08-11

**Soundness:** 2

**Excitement:**

3: Ambivalent: It has merits (e.g., it reports state-of-the-art results, the idea is nice), but there are key weaknesses (e.g., it describes incremental work), and it can significantly benefit from another round of revision. However, I won't object to accepting it if my co-reviewers champion it.

**Paper Topic And Main Contributions:**

The work proposes a method for the application of accent transfer (F0 and rhythm) between two different speakers and accents. The evaluation is performed on two different datasets, the VCTK corpus and a Lao corpus with no parallel-corpus (same speaker with different accents). The proposed method mixes most of the ideas from the MaskCycleGAN-VC (Kaneko et al. 2021) and combines the mutual information learning from SRD method (Yang et al. 2022), adding the ability of the model to segregate speaker features, accent features and sentence's content. The method also improves the methods from (Kaneko et al. 2021) by adding feature disentanglement components (C_1, C_2) to classify speakers by using speaker features or accent features, respectively.

**Questions For The Authors:**

- QUESTION A: Is the E_ac part trained independently for each dataset (Engllish, Lao) or are different depending on experiments?
- QUESTION B: The same question arises for the E_s and E_r modules? are they retrained for Lao? are you using same English models?
- QUESTION C: The Lao dataset is a self-constructed dataset so it lacks a reference, is it possible to make it available?
- QUESTION D: what is z_p in the figure 1? is it the output of the E_ac?
- QUESTION E: For the vCLUB in the adversarial classifier C_2, what is the q_\tetha in the equation (6)? and the indexes i and j?
- QUESTION F: How the results from table 1, where SRD seems to be similar or even better (in SSIM score table 1) relates to the figure 3 where feature confusion looks bigger for SRD? are the same Engilsh speakers involved in both reportings?

**Reasons To Accept:**

- The approach and ideas seem interesting enough and use state of the art methods to address a challenging and novel task as it is speaker accent transfer.
- The experiments are conducted on two different datasets and the results obtained in the low-resource scenario are above sota.

**Reasons To Reject:**

- The training methodology is not well explained and is hard to follow.
- The section 2.4. is showing the addition of several loss from each of the components. To me is not clear what L function stands for and most of the different losses are not explained in the text or some weighting parameters as alpha, beta, gamma or lambda, the latter seeming to control the cycle content loss and the speaker id loss.
- It is also hard to understand the MI loss obtained from the C_2 discriminator.
- Low number of speakers are reported in the tables for English.

**Reproducibility:**

4: Could mostly reproduce the results, but there may be some variation because of sample variance or minor variations in their interpretation of the protocol or method.

**Reviewer Confidence:**

3: Pretty sure, but there's a chance I missed something. Although I have a good feel for this area in general, I did not carefully check the paper's details, e.g., the math, experimental design, or novelty.

**Typos Grammar Style And Presentation Improvements:**

- The reference (Mohsen and Nirmal) is not well chosen and other references will benefit even more the paragraph.
- In line 46, concatenate all references within the same parenthesis.
- In lines 79-80: you could remove "these fine-grained features" to make shorter and a more readable sentence.
- In line 119, "an accent feature..."
- In line 152: "a speach content"
- In Eq(1), the first line: S --> E_r(S)
- In line 191, row --> raw
- It is hard to follow the text and relate it to the figure 1. Such figure could be improved by, e.g. , a dashed square on the encoder content part, adding E_c. Also adding E_ac in the accent encoder dashed square. Also I would suggest add the references (Chen, 2021) within the E_s box and (Quian, 2020) in the E_r one. To make it consistent with the main text, the output "Converted Spectogram (\hat S)
- In line 332: "hiiden"
- In line 342: "selecte 200"
- In the line 380: the method is not outperforming SRD in SSIM scoring.
- In the paragraph (lines 383-388): Add a reference to table 1.
- To improve the contrast in figure 6, I would suggest to switch the color for the F0 contours.
- In section 3.3.4. line 456-457: the sentence "but there are significant differences in the contour of the F0 pitch" is saying the contrary that 6 lines above.

---

> ### Author Rebuttal · Authors · 2023-08-28
>
> (1). The model training strategy adopts a non-parallel voice conversion approach. Given a non-parallel corpus D(x,y), the training mechanism involves mapping x to y and then back to x', with the primary training objective being the minimization of the mean square error between x and x'. The distinctiveness of this research lies in its innovation of incorporating accent feature modeling and disentanglement. During the model inference phase, various latent variables corresponding to different accents are selectively employed in model decoding. This enables the controlled manipulation of accent transfer, serving as a pivotal contribution of this paper.
> (2). Here is a detailed explanation of each L loss function
>  $\mathcal{L}_{cyc}^{X\longrightarrow Y\longrightarrow X} =\mathbb{E}_x~P_X[||G_{Y\to X}(G_{X\to Y}(x))-x||_1]$,
>  $\mathcal{L}_{id}^{X\longrightarrow Y} =\mathbb{E}_{y\sim P_Y}[||G_{X\to Y}(y)-y||_1]$,
>  \mathcal{L}_{adv2}^{X\longrightarrow Y\longrightarrow Z} =\mathbb{E}_{x\sim P_X}[\log_{}{D'_X}(x)]
> +\mathbb{E}_{x\sim P_X}[\log_{}{(1-D'_X}(G_{Y\to X}(G_{X\to Y}(x))))],
> where x represents S_w(A_w), y represents S_h(A_h), G is the generator，D is the discriminator.
> The alpha, beta, gamma or lambda are set 0.1 in the experiments.
> (3). The MI loss is not obtained from the C_2 discriminator, which is minimizing the unbiased estimator for vCLUB. The C_2 is designed to gradually struggle in correctly classifying the speaker ID for Z_{ac} and Z_c. And \mathcal{L}_{adv-C_2} is the C_2 loss by using adversarial speaker classifier with GRL.
> (4). We intend to augment the experimentation involving English speakers in Table 1 by incorporating a minimum of 4 English speakers characterized by distinct accent migrations.
> (5).
> QUESTION A: Is the E_ac part trained independently for each dataset (Engllish, Lao) or are different depending on experiments?
> ANSWER A: The E_ac part is independently for each dataset，is depending on language.
>
> QUESTION B: The same question arises for the E_s and E_r modules? are they retrained for Lao? are you using same English models?
> ANSWER B: The E_s and E_r modules independently for each dataset, and they are retrained for Lao.
>
> QUESTION C: The Lao dataset is a self-constructed dataset so it lacks a reference, is it possible to make it available?
> ANSWER C: We'll be open-sourcing the code and the self-constructed dataset.
>
> QUESTION D: what is z_p in the figure 1? is it the output of the E_ac?
> ANSWER D: z_p is the output of E_{VQ-VAE}(P)), Z_p \oplus {Z_r} is the output of the E_ac.
>
> QUESTION E: For the vCLUB in the adversarial classifier C_2, what is the q_\tetha in the equation (6)? and the indexes i and j?
> ANSWER E: q_\tetha(Z_ac |Z_c) is a variational distribution with parameter \tetha to approximate p(Z_ac |Z_c). The unbiased estimator for vCLUB with samples {Z_{{ac}_i, Z_c_i}. The indexes i and j are the samples of Z_ac and Z_c.
>
> QUESTION F: How the results from table 1, where SRD seems to be similar or even better (in SSIM score table 1) relates to the figure 3 where feature confusion looks bigger for SRD? are the same Engilsh speakers involved in both reportings?
> ANSWER F: The SSIM scores presented in Table 1 offer only a partial representation of the situation. The similarity between the SSIM scores of the SRD model and our proposed model, and in some cases, even the superiority of SRD's SSIM scores, can be attributed to the following reason: In the SSIM calculation of S_{p248}(A_{p248}) \Longleftrightarrow S_{p329}(A_{p329})}, a step-wise procedure is followed. Initially, the SSIM calculation of S_{p248}(A_{p248}) \Longleftrightarrow S_{p329}(A_{p329})}\Longleftrightarrow {\hatS}_{p248}(A_{p248})} is performed, and subsequently, the SSIM calculation of S_{p248}(A_{p248}) and {\hatS}_{p248}(A_{p248})} is executed. This involves an x->y->x' transformation process, the SSIM is the distinctions of x and x'. As illustrated in Figure 3. Consequently, even when the features are not entirely disentangled, the SRD model can still achieve commendable performance. However, it's vital to note that this does not align with the primary objective of this paper, which revolves around accent migration.

---

### Meta-Review · Area_Chair_JaTh · 2023-09-14

**Recommendation:** 3

**Metareview:**

This paper looks at the challenging problem of converting speech so that it appears to be produced by a speaker with a different accent from the source. The paper looks specifically at pitch and rhythm modelling. The reviewers agreed that the good performance of the proposed approach to a relevant task was a strength. But they also all highlighted that parts of the paper were difficult to follow, especially with regard to missing technical details. Although the authors provided follow-up explanations in the rebuttal period, this did not alter the reviewers' opinion regarding the clarity. Nevertheless, despite ambivalent excitement, the reviewers seem to agree that the work is generally sound.

---

### Decision · Program_Chairs · 2023-10-07

**Decision:**

Accept-Findings

**Comment:**

This paper looks at the challenging problem of converting speech so that it appears to be produced by a speaker with a different accent from the source. The paper looks specifically at pitch and rhythm modelling. The reviewers agreed that the good performance of the proposed approach to a relevant task was a strength. But they also all highlighted that parts of the paper were difficult to follow, especially with regard to missing technical details. Although the authors provided follow-up explanations in the rebuttal period, this did not alter the reviewers' opinion regarding the clarity. Nevertheless, despite ambivalent excitement, the reviewers seem to agree that the work is generally sound.